

# SPCANet: congested crowd counting *via* strip pooling combined attention network

Zhongyuan Yuan

College of Information and Intelligence, Hunan Agricultural University, Changsha, Hunan Province, China

Corresponding author
Zhongyuan Yuan,
1289994390@qq.com

## ABSTRACT

Crowd counting aims to estimate the number and distribution of the population in crowded places, which is an important research direction in object counting. It is widely used in public place management, crowd behavior analysis, and other scenarios, showing its robust practicality. In recent years, crowd-counting technology has been developing rapidly. However, in highly crowded and noisy scenes, the counting effect of most models is still seriously affected by the distortion of view angle, dense occlusion, and inconsistent crowd distribution. Perspective distortion causes crowds to appear in different sizes and shapes in the image, and dense occlusion and inconsistent crowd distributions result in parts of the crowd not being captured completely. This ultimately results in the imperfect capture of spatial information in the model. To solve such problems, we propose a strip pooling combined attention (SPCANet) network model based on normed-deformable convolution (NDConv). We model long-distance dependencies more efficiently by introducing strip pooling. In contrast to traditional square kernel pooling, strip pooling uses long and narrow kernels ($1 \times N$ or $N \times 1$) to deal with dense crowds, mutual occlusion, and overlap. Efficient channel attention (ECA), a mechanism for learning channel attention using a local cross-channel interaction strategy, is also introduced in SPCANet. This module generates channel attention through a fast 1D convolution to reduce model complexity while improving performance as much as possible. Four mainstream datasets, Shanghai Tech Part A, Shanghai Tech Part B, UCF-QNRF, and UCF CC 50, were utilized in extensive experiments, and mean absolute error (MAE) exceeds the baseline, which is 60.9, 7.3, 90.8, and 161.1, validating the effectiveness of SPCANet. Meanwhile, mean squared error (MSE) decreases by 5.7% on average over the four datasets, and the robustness is greatly improved.

## INTRODUCTION

Crowd counting is a vital landing direction for computer vision. Crowd counting involves estimating the number, density, and distribution of people in an image (*Chen et al., 2013*; *Lempitsky & Zisserman, 2010*; *Zhang et al., 2016*) or video (*Ge & Collins, 2009*; *Chen, Fern & Todorovic, 2015*). It is a core issue, and research focuses on intelligent video surveillance. With the global population explosion and accelerated urbanization, high-density gatherings have become more frequent. Coupled with improper site management, this can quickly lead

to crowd stampede accidents (*Abdelghany et al., 2014*; *Almeida, Rosseti & Coelho, 2013*), such as the Itaewon stampede on October 29, 2022, in Seoul, South Korea (*Ha, 2023*). The importance of crowd counting is becoming more and more prominent. Good counting algorithms can also be extended to other related fields, such as traffic management to achieve better flow control (*Guerrero-Gómez-Olmedo et al., 2015*) and ecological detection to protect the environmental balance. Therefore, studying crowd-counting methods has important practical significance and application value.

Currently, there are two main mainstream counting methods, tracking-based methods (*Dollar et al., 2011*; *Topkaya, Erdogan & Porikli, 2014*; *Li et al., 2008*) and feature regression-based methods (*Chen et al., 2012*; *Chan & Vasconcelos, 2009*), which have certain limitations and cannot accomplish the task well in the face of complex scenes, dense crowds, and mutual occlusion, *etc*. They are gradually replaced by convolutional neural network (CNN)-based methods (*Wang et al., 2015*; *Zhang et al., 2016*; *Sam, Surya & Babu, 2017*; *Sindagi & Patel, 2017*; *Li, Zhang & Chen, 2018*), which can extract features from raw image data through end-to-end learning and make counting predictions without manually designing features or complex pre-processing. This makes the model more adaptive and generalized. Crowd counting tasks usually involve complex scenes, such as crowd density, occlusion, overlapping, *etc*. CNN can automatically learn and adapt to these complex scenes when processing images, while tracking and regression methods may require more manual adjustments and scene-specific designs. CNNs perform better when dealing with scale-varying and cross-scene problems, and in comparison, CNNs are far superior to their predecessors in terms of performance, efficiency, and robustness. Thus, they have become the focus of research nowadays. Although CNN-based crowd-counting solutions have achieved remarkable results, it has been found that simply increasing the depth of the network model brings new problems. The network model has many parameters, making it difficult to train; the multi-branch structure learns features with high similarity, *etc*. These constraints limit the improvement of counting accuracy.

To address the aforementioned challenges, strategies such as modifying the receptive field have been commonly adopted (*Li, Zhang & Chen, 2018*; *Dai et al., 2017*; *Zhu et al., 2019*; *Zhong et al., 2022*). *Li, Zhang & Chen (2018)* utilized the dilated convolution technique to expand the receptive field, preserving image resolution and thus retaining more detailed image information. *Zhu et al. (2019)* introduced a deformable convolution (deformable conv) that adaptively adjusts the receptive field based on the input data's shape characteristics, minimizing the information loss inherent in traditional convolution methods. Furthermore, *Zhong et al. (2022)* developed a novel approach, normed-deformable convolution (NDConv), powered by normed deformable loss (NDloss). This method enhances the network through a geometric transformation operator, where the offset is controlled by NDConv, enabling a more comprehensive capture of head features for uniform head sampling. Improving the quality of density map generation has also been a focus, with optimization of loss functions playing a pivotal role (*Cao et al., 2018b*; *Goodfellow et al., 2014*; *Ma et al., 2019*). Scale aggression network (SANet) assesses density map quality using an image quality evaluation criterion, the Structural Similarity Index (SSIM), as proposed by *Cao et al. (2018b)*. Meanwhile, general adversarial networks

(GANs) have been employed to produce high-quality density maps through adversarial loss derived from a continuous interaction between the generative and discriminative networks. The Bayesian loss approach also models the density contribution probability from annotated points, comparing this probability with the estimated density at each pixel. The multiplication of the contribution probability with the estimated density for each pixel is aggregated to compute the expected count for each annotation point.

However, the aforementioned techniques often struggle with accurately capturing spatial information, hindered by challenges such as perspective distortion, dense occlusion, and inconsistent crowd distribution. These factors significantly impair the precision of model-based counts, especially with varying crowd densities and obstacles. To solve the above problems, this article introduces strip pooling (*Hou et al., 2020*), on top of the basic model architecture of NDConv, as a novel spatial pooling strategy to cope with the aberration problem caused by the distortion of viewpoints and the diversity of crowd distribution. Compared with traditional square kernel pooling, strip pooling uses long and narrow kernels ($1 \times N$ or $N \times 1$) to model long-range dependencies more efficiently and is suitable for dealing with dense crowds, mutual occlusion, and overlapping. By introducing the strip pooling module, this article aims to improve the ability to model human-body boundary relationships, thus enhancing crowd counting in highly crowded and noisy scenes. In addition, since strip pooling is designed as a lightweight and efficient spatial pooling strategy, its introduction does not cause a significant increase in the computational burden of the model, which helps to improve the computational efficiency in large-scale crowd scenarios. Since the strip pooling module is designed for pixel-level prediction tasks, its introduction into the crowd-counting network can make the network more adapted to handle data with pixel-level annotations, which will be more advantageous than the original model in dense scenarios. Additionally, this study integrates the efficient channel attention module (ECA) (*Wang et al., 2020*), which uses quick 1D convolution to generate channel attention. ECA significantly boosts performance with minimal parameter involvement as a streamlined channel attention mechanism, underscoring the model's enhanced efficiency and effectiveness in crowd-counting endeavors.

To summarize, this article makes three main contributions.

(1) Compared with traditional neural networks, strip pooling combined attention (SPCANet) has introduced a novel spatial pooling strategy, the strip pooling module, which makes the backbone network more effective in modeling long-distance dependencies in a lightweight yet efficient way.

(2) SPCANet introduced a channel attention module (ECA) into the model to maximize model performance with few parameters.

(3) SPCANet incorporates a simple yet effective spatial pooling strategy and an attention mechanism that outperforms other methods on four datasets for dense scene counting, providing new perspectives and solutions to the task of dense scene counting.

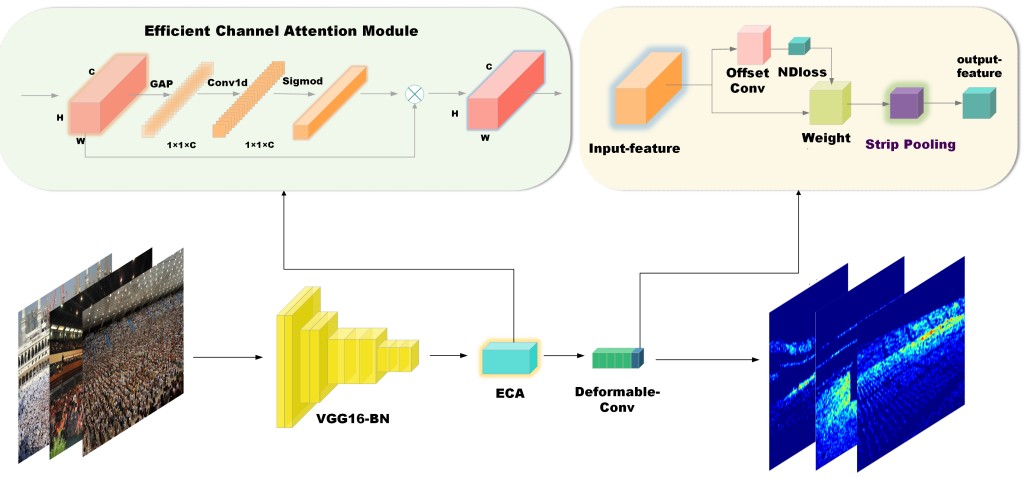

**Figure 1** Structure of the SPCANet.

# RELATED WORK

## Detection-based methods

Early research on crowd counting focused on detection-based methods that use sliding windows (*Dollar et al., 2011*) to detect pedestrians or ecological parts of a scene and count the corresponding number of people. Detection-based methods are mainly categorized into whole-based detection and part-body-based detection. Whole-based detection methods extract low-level behavioral features from the whole of a person, such as Haar wavelets (*Viola & Jones, 2004*) and histogram-oriented gradients (HOG) (*Dalal & Triggs, 2005*), and the learning algorithms for the classifiers are mainly support vector machine (SVM), boosting, and Random Forest methods. Since whole-based detection methods are only suitable for sparse populations, methods based on partial body detection have been used to deal with denser population counting problems. The detection is mainly performed by localized body structures such as head, arms, *etc.* (*Felzenszwalb et al., 2009*; *Wu & Nevatia, 2007*); the results are slightly improved compared to the overall detection. Overall, the detection-based approach performs well in sparse crowd-counting tasks but is not applicable in dense scenes with multi-scale and severe occlusion.

## Regression-based methods

When detection-based methods cannot satisfy the counting task in dense scenes, researchers began to accomplish counting by extracting low-level features from localized image blocks and then establishing mapping relationships with regression models. The main idea is to learn a mapping from a feature to a crowd count (*Chan & Vasconcelos, 2009*; *Ryan et al., 2009*; *Chen et al., 2012*). The low-level features include foreground features, edge features (*Chan, Liang & Vasconcelos, 2008*; *Chan & Vasconcelos, 2011*; *Chen et al., 2012*), texture, and gradient features; the regression models are mainly, for example, linear regression (*Paragios & Ramesh, 2001*), segmented linear regression (*Chan, Liang & Vasconcelos, 2008*), ridge regression (*McDonald, 2009*), and Gaussian process regression.

The regression method effectively alleviates the limitations in high-density scenes when based on the detection method and eliminates the dependence on detectors. The counting in dense scenes has been substantially improved, but at the same time, it exacerbates the computational complexity.

## CNN-based methods

In recent years, convolutional networks have demonstrated powerful capabilities in deep-level feature extraction in images, so they have been widely used in crowd-counting tasks. *Wang et al. (2015)* introduced CNN into crowd counting for the first time and constructed an end-to-end CNN regression model. *Zhang et al. (2016)* proposed a multicolumn convolutional neural network (MCNN) to solve the multi-scale problem in 2016 and introduced the crowd-counting classical dataset, Shanghai tech. In 2017, *Sam, Surya & Babu (2017)* improved on the multicolumn convolutional neural network and proposed SWITCH-CNN, which improves the model's ability to adapt under drastic scale changes. In the same year, *Sindagi & Patel (2017)* proposed a context-aware counting network CP-CNN to enhance the quality of generated density maps.

In 2018, *Li, Zhang & Chen (2018)* proposed the dilated convolutional neural network CSRNet, which for the first time applies dilated convolution to the model, enlarges the sensory field, better understands highly congested scenarios, performs accurate counting estimation, and provides high-quality density maps. *Cao et al. (2018a)* proposed a novel encoder–decoder network, called scale aggregation network (SANet), to generate high-quality density maps.

In 2019, *Liu et al. (2019)* proposed a variable convolutional network for crowd understanding attention mapping, introducing an attention mechanism to emphasize crowd location, achieving the ability to capture crowd features more efficiently and with more robust resistance to various noises. In the same year, *Zhong et al. (2022)* proposed a new convolution network, which is capable of handling continuous scale variations of individual pedestrians. With spatially varying Gaussian smoothing, the perspective guided convolution (PGC) can adaptively use different sizes of receptive fields for various scales of people. In the same starting point as PGC, context-aware network, proposed by *Liu, Salzmann & Fua (2019)*, captures contextual information at different scales using multi-scale feature pyramids. Each scale corresponds to a different receptive field size, and then the features are combined and passed to subsequent convolutional layers.

Recently, *Cheng et al. (2022)* proposed a network structure called GauNet, which suggests a low-rank approximation with translation invariance to realize well the approximation of large-scale Gaussian convolution to speed up the computation. *Ma, Zhang & Wei (2024)* introduced an end-to-end model, FGENet, for precise human pose estimation. Unlike other methods that estimate density maps, FGENet directly learns the original coordinate points, minimizing the impact of labeling noise. The model performs better than previous methods on various datasets. Table 1 lists the models and features that have emerged in recent years.

**Table 1 Overview of mainstream models.**

| Network structure | Representative model | Advantage | Shortcoming | Dataset | Result (MAE) |
|---|---|---|---|---|---|
| Single branch structure | CrowdCNN | Can handle scenes with varying densities. | Limited adaptability to density variations. | WorldExpo'10 | 12.9 |
| Multi-branch structure | MCNN | Multi-column architecture can handle various density changes. | High number of parameters and complex training. | ShanghaiTechPartB | 26.4 |
| | CP-CNN | Combines contextual information, leading to high accuracy. | High computational complexity. | ShanghaiTechPartB | 20.1 |
| | Switch-CNN | Dynamically switches sub-networks to handle different density scenes. | Complex switching mechanism and difficult training. | ShanghaiTechPartA | 90.4 |
| Special structure | CSRNet | Uses dilated convolutions to efficiently capture multi-scale information. | Highly dependent on training data. | ShanghaiTechPartA | 68.2 |
| | ACSCP | Adaptive scheme, effective in high-density crowd scenarios. | Poor real-time performance. | UCF_CC_50 | 291.0 |
| | GauNet | Uses Gaussian filters to improve density estimation accuracy. | Sensitive to outliers, less robust. | UCF_CC_50 | 186.3 |

## METHODS

This section first introduces the model architecture, then reviews NDloss (*Zhong et al., 2022*), introduces strip pooling, and finally, efficient channel attention (ECA).

### Net architecture

In designing the model architecture, we focused on the following three aspects:

(1) Feature hierarchy: Strip pooling and ECA modules are typically used to introduce higher-level feature representations and capture global information in deep networks. Therefore, placing them in the deeper layers of the model can better utilize deep features for feature enhancement and global attention.

(2) Information fusion: The strip pooling module enhances the model's ability to perceive information at different scales through multi-scale pooling and feature fusion. For the original model, the NDloss module effectively aggregates head information. Placing the strip pooling module immediately after the NDloss module allows for better information fusion and pooling operations at higher-level feature representations. The primary function of the ECAAttention module is to enhance the model's focus on essential features while suppressing unimportant ones. Placing it in the deeper layers leverages global information to adjust feature weights more effectively, enhancing the model's feature representation capability. We put it just before the final feature output.

(3) Computational efficiency: Integrating these modules at deeper layers inevitably increases computational load. It is crucial to effectively utilize lower-level feature representations while maintaining computational efficiency. In the ablation experiments, we discussed the impact of the number of strip pooling layers on model performance and concluded that adding strip pooling after two NDConv layers yields the best results.

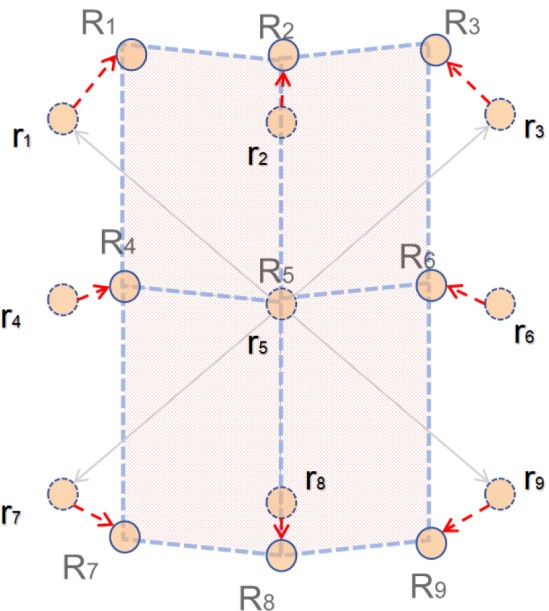

**Figure 2** Diagram of NDloss.

As shown in Fig. 1, our model is built on top of NDConv. Compared to CSRNet, the authors added a batch normalization layer after each convolutional layer for enhanced training robustness on cropped images. They replaced the last null convolution layer with NDConv, creating a model (backbone) denoted as NDConv. We improve on this by adding an ECA module with kernel size = 3 between the original model VGG16_BN (frontend) and deformable-conv (backend), which improves the model's ability to sense between channels and involves very few parameters. Between the backend and the final output_layer, a new lightweight strip pooling module with 64 input channels and a pooling size of (20, 12) is inserted to enable the backbone network to model remote dependencies efficiently.

Additionally, in choosing the backbone, we considered the simplicity of the VGG16 network structure, which mainly consists of convolutional layers, max-pooling layers, and fully connected layers. This simple structure makes it easy to understand and implement for crowd-counting tasks. VGG16 achieved excellent results in the ImageNet competition, demonstrating its strong performance in image classification tasks. Since crowd counting also involves complex feature extraction and classification problems, VGG16's outstanding performance makes it an ideal choice. In recent years, VGG16 has been particularly favored by researchers in the field of crowd counting, as evidenced by studies such as CSRNet (*Li, Zhang & Chen, 2018*) and context-aware network (CAN) (*Liu, Salzmann & Fua, 2019*). We used VGG16_BN instead of VGG16 as the backbone. Compared to VGG16, VGG16_BN incorporates batch normalization, which makes the network more stable, speeds up convergence, and effectively prevents overfitting.

## Normed-deformable loss

As shown in Fig. 2, *Zhong et al. (2022)* used four parallelograms $R_1R_2R_5R_4$, $R_2R_3R_6R_4$, $R_4R_5R_8R_7$, $R_5R_6R_9R_8$ to simplify the prior. We know that if for deformable convolution, the sampling point will correspondingly move around r1–r9 due to the offset. This case involves two-dimensional offsets, expressed as follows: $(\Delta R_{1x}, \Delta R_{1y}), \cdots, (\Delta R_{9x}, \Delta R_{9y})$, where we have $R_1 = \Delta R_{1x} + r_1, \cdots, R_9 = \Delta R_{9x} + r_9$. Next, we use the simplified parallelogram to restrict the sampling points and construct the corresponding losses. The restricted sampling points should try to satisfy the following three conditions: (1) The center sampling point $R_5$ should be close to $r_5$ as a matter of course. (2) $R_4$ and $R_6$ are at the same distance from $r_5$ and should be close to the $x$-axis. (3) Similarly $R_2$ and $R_8$ should have equal distances from $r_5$ and both should be close to the $y$-axis.

For condition Eq. (1), it follows that:

$$\mathcal{L}_{r_5} = \left\| \Delta R_{5x} \right\|_2^2 + \left\| \Delta R_{5y} \right\|_2^2 \tag{1}$$

For condition Eq. (2), it follows that:

$$\mathcal{L}_{hor} = \left\| \Delta R_{4x} + \Delta R_{6x} \right\|_2^2 + \left\| \Delta R_{4y} \right\|_2^2 + \left\| \Delta R_{6y} \right\|_2^2 \tag{2}$$

For condition Eq. (3), it follows that:

$$\mathcal{L}_{vec} = \left\| \Delta R_{2y} + \Delta R_{8y} \right\|_2^2 + \left\| \Delta R_{2x} \right\|_2^2 + \left\| \Delta R_{8x} \right\|_2^2 \tag{3}$$

$$\begin{aligned}
\mathcal{L}_{R_1} &= \left\| (r_4 + \Delta R_4) + (r_2 + \Delta R_2) - (r_5 + \Delta R_5) - r_1 \right\|_2^2 \\
\mathcal{L}_{R_3} &= \left\| (r_4 + \Delta R_4) + (r_2 + \Delta R_2) - (r_5 + \Delta R_5) - r_1 \right\|_2^2 \\
\mathcal{L}_{R_7} &= \left\| (r_4 + \Delta R_4) + (r_8 + \Delta R_8) - (r_5 + \Delta R_5) - r_6 \right\|_2^2 \\
\mathcal{L}_{R_9} &= \left\| (r_6 + \Delta R_6) + (r_8 + \Delta R_8) - (r_5 + \Delta R_5) - r_9 \right\|_2^2
\end{aligned} \tag{4}$$

Then, NDLoss can be expressed as:

$$\mathcal{L}_{nd} = \mathcal{L}_e + \mathcal{L}_{hor} + \mathcal{L}_{vec} + \mathcal{L}_{R_1} + \mathcal{L}_{R_3} + \mathcal{L}_{R_6} + \mathcal{L}_{R_9} \tag{5}$$

Then, the final loss is:

$$\mathcal{L}_{all} = \mathcal{L}_{den} + \lambda \mathcal{L}_{nd} \tag{6}$$

where $\lambda$ is the super-parameter. The constraint of normed-deformable loss (*i.e.*, NDLoss) makes the sampling point offsets in the deformable convolution more uniform, allowing better access to head features, effectively improving the sensory field conditioning, and improving the model performance. In addition, NDLoss is a lightweight module that does not add to the computational burden.

## Strip pooling

Strip pooling (*Hou et al., 2020*) is a spatial pooling method that is very effective when dealing with complex target scenes and especially excels when focusing on long-range contextual information. Compared to traditional pooling operations, it has the advantage

of reducing the impact of including disjoint regions. The method employs a strip pooling window to operate along the horizontal or vertical direction. For a tensor $\mathbf{x} \in \mathbb{R}^{H \times W}$, in the average pooling layer, the spatial extent of (h x w) is pooled. The output y is also a two-dimensional tensor, with height $H_o = \frac{H}{h}$ and width $W_o = \frac{W}{w}$. The pooling operation can be written as follows:

$$y_{i_o,j_o} = \frac{1}{h \times w} \sum_{0 \leq i < h} \sum_{0 \leq j < w} x_{i_o \times h + i, j_o \times w + j} \tag{7}$$

where $0 \leq i_o < H_o$ and $0 \leq j_o < W_o$. Strip pooling differs from average pooling by pooling the spatial extent of (H,1) or (1, W) and averaging all feature values in a row or column. The output $\mathbf{y}^h \in \mathbb{R}^H$ after horizontal strip pooling is calculated by averaging all values in a row:

$$y_i^h = \frac{1}{W} \sum_{0 \leq j < W} x_{i,j} \tag{8}$$

Vertical strip pooling output $\mathbf{y}^v \in \mathbb{R}^w$ can be written in a compact form:

$$y_j^v = \frac{1}{H} \sum_{0 \leq i < H} x_{i,j} \tag{9}$$

By introducing horizontal and vertical banded pooling layers, long-term relational dependencies between discretely distributed regions across the dataset can be easily identified and maintained. In particular, this approach performs well when dealing with areas with banded coding due to the unique elongated structure of the employed kernel. The kernel's elongated shape helps maintain focus, thus better isolating and capturing localized complexities and features.

The implementation process of the strip pooling algorithm is not complex. It captures and fuses multi-scale and multi-directional features through a series of feature extraction, pooling, and convolution operations. Initially, two $1 \times 1$ convolution layers are defined to preliminarily process the input feature map, generating two intermediate feature maps. One of these intermediate feature maps undergoes three different sizes of adaptive pooling, followed by $3 \times 3$ convolutions and upsampling back to the original size. Simultaneously, the other feature map undergoes horizontal and vertical pooling separately, followed by convolution and upsampling to the original size. Subsequently, these processed feature maps are additively fused and further processed through a $3 \times 3$ convolution layer. Finally, the fused feature maps are concatenated along the channel dimension, restored to the original input channel count *via* a $1 \times 1$ convolution layer, and added to the input feature map. The result is activated using the ReLU activation function to generate the final output. By employing this multi-scale and multi-directional feature processing and fusion, the strip pooling module effectively enhances the model's feature representation capability in complex scenarios.

The improvements to this module primarily focus on three aspects:

(1) Enhancement of pooling operations: The original article mentions pool sizes of (20, 12) and (12, 20). In the implementation, the dimensions of pool1 and pool2 vary according

to the different cropping sizes of images from different datasets. This modification is mainly to meet the requirements of feature extraction in specific scenarios. Additionally, pool3 and pool4 have been added with pooling sizes of (1, None) and (None, 1), respectively, to capture as many as possible features in the horizontal and vertical directions.

(2) Improvement in feature fusion: The configuration of the conv2_x series convolution layers has been modified, using convolution kernels of (1, 3) and (3, 1), in contrast to the conventional (3, 3) convolution.

(3) Other details: Bilinear mode was selected for upsampling with align_corners set to true. This helps preserve spatial information better during the upsampling process.

### Efficient channel attention

Efficient channel attention (ECA) (*Wang et al., 2020*) is a lightweight channel attention module for improving the performance of convolutional neural networks (CNNs). ECA uses a strategy called local cross-channel interaction, which effectively avoids the impact of dimensionality reduction on the learning effect of channel attention. The block involves only a few parameters but has a significant effect gain. The original article compares three variants of the squeeze-and-excitation (SE) block (*Hu, Shen & Sun, 2018*), SE-Var1-3, in which SE-Var3 is more effective than SE-Var2 because SE-Var3 captures inter-channel interactions while SE-Var2 does not. The author explores another method to capture local cross-channel interaction in ECA, using a band matrix $\mathbf{W}_k$ to learn channel attention.

$$
\begin{bmatrix}
w^{1,1} & \cdots & w^{1,k} & 0 & 0 & \cdots & \cdots & 0 \\
0 & w^{2,2} & \cdots & w^{2,k+1} & 0 & \cdots & \cdots & 0 \\
\vdots & \vdots & \vdots & \vdots & \ddots & \vdots & \vdots & \vdots \\
0 & \cdots & 0 & 0 & \cdots & w^{C,C-k+1} & \cdots & w^{C,C}
\end{bmatrix}
\tag{10}
$$

In Eq. (10), only interactions between $y_i$ and its $k$ neighbors are considered in the weights, *i.e.*,

$$
\omega_i = \sigma\left(\sum_{j=1}^{k} w_i^j y_i^j\right), y_i^j \in \Omega_i^k,
\tag{11}
$$

This strategy can be achieved by a fast $1D$ convolution with kernel size $k$, *i.e.*,

$$
\omega = \sigma(\text{C1D}_k(y)),
\tag{12}
$$

where C1D stands for $1D$ convolution. In particular, the ECA module exhibits similar results to SE-var3 at $k = 3$ but with lower complexity. It efficiently captures local cross-channel interactions, ensuring high efficiency and effectiveness. The channel attention mechanism has great potential to improve the performance of counting CNNs by dynamically adjusting the weights of each channel based on the correlation between different channels to enhance the model's feature extraction and exploitation in extreme environments such as distorted viewpoints and dense occlusion between crowds. However, the complex attention mechanism module inevitably increases the complexity of the model. The introduction of ECA successfully achieves a good balance, is lightweight, and is fast to help the model learn effective channel attention.

As a lightweight channel attention mechanism, ECA defines an adaptive average pooling layer, a 1D convolution layer, and a Sigmoid activation function during the initialization phase for global feature capture and attention weight generation. In the forward propagation, the module first performs global average pooling on the input feature map using the adaptive average pooling layer. It then captures global correlations through the 1D convolution layer, followed by mapping the obtained attention weights to a range between 0 and 1 using the Sigmoid activation function. Finally, the module performs an element-wise multiplication of the attention weights with the input feature map, resulting in a weighted feature representation. This process enhances the model's focus on essential features, thereby improving the model's feature representation capability.

As a lightweight channel attention module, there were no significant modifications when embedding it into the model; the focus was mainly on its placement. After extensive testing, we found that adding this module after VGG16_BN, specifically after the front end, resulted in the most significant performance improvement. A possible theoretical explanation lies in the net architecture.

## EXPERIMENT

In this section, we first briefly introduce experiment settings and the evaluation metrics and datasets used, then compare the results of our model on the dataset with current state-of-the-art models, and finally perform an ablation study to find the contribution made by each new improvement in the model.

### Experiment settings

The mean squared error loss (MSELoss) was used to supervise the training of SPCANet, with an Adam optimizer with a learning rate of 1e−4, batch size of 4, a decay rate of 0.9, and momentum of 0.95 for optimization. Experiments were performed using Pytorch (1.11.0+cu113) and on a GeForce RTX 3090 graphics card. The pseudo-code for model training is shown in Algorithm 1.

### Evaluation metrics

We use mean absolute error (MAE) and mean squared error (MSE) to assess model accuracy and robustness. MAE and MSE are specifically defined for a sample size of N as:

$$MAE = \frac{1}{N}\sum_{i=1}^{N}|R_i - R_i^{\mathrm{GT}}| \tag{13}$$

and

$$MSE = \frac{1}{N}\sum_{i=1}^{N}(R_i - R_i^{\mathrm{GT}})^2 \tag{14}$$

where $R_i$ denotes the number of the population predicted by the model and $R_i^{\mathrm{GT}}$ denotes the number of the real population. MAE can effectively reflect the model's accuracy, and MSE can reflect the model's robustness. The combination of the two can comprehensively evaluate the model's performance.

---

**Algorithm 1** SPCANet training algorithm

---

**Input:** $N$ training image patches $\{X_i\}_{i=1}^N$ with groundtruth density maps $\{D_{X_i}^{GT}\}_{i=1}^N$

**Output:** Trained parameters $\{\Theta_k^i\}_{k=1}^{Td}$ for $R_k$

/*Training for $E_p$ epochs*/
/*$r_i^k$ is result predicted by $R_k$ for input $X_i$*/
/*$r_i^{GT}$ is ground truth count for input $X_i$*/
/*$O_i$ is NDloss offset matrix*/
**for** $e = 1$ to $E_p$ epochs **do**
    Save model with $R_e$ epoch and MAE;
    **for** $i = 1$ to $N$ **do**
        /*count with SPCANet*/
        $r_i^k, O_i = \text{model}(X_i)$;
        **for** $j = 1$ to $O_i$ **do**
            NDloss generates *extra_loss_i*;
        **end for**
        $loss_i^{best} = \underset{k}{\text{argmin}}|r_i^k - r_i^{GT}|$;
        optimizer.zero_grad()
        $(loss_i^{best} + extra\_loss_i)$.backward()
    **end for**
**end for**

---

**Table 2** Overview of the four datasets.

| Dataset | Resolution | Number of pictures | Number of training images | Number of test images | Maximum count | Minimum count | Average count | Total count |
|---|---|---|---|---|---|---|---|---|
| Shanghai Tech Part A | Different | 482 | 300 | 182 | 3,139 | 33 | 501.4 | 241,677 |
| Shanghai Tech Part B | 768 × 1,024 | 716 | 400 | 316 | 578 | 9 | 123.6 | 88,488 |
| UCF_CC_50 | Different | 50 | 40 | 10 | 4,543 | 94 | 1,279.5 | 63,974 |
| UCF-QNRF | Different | 1,535 | 1,201 | 334 | 12,865 | 49 | 815.4 | 1,2511,642 |

## Datasets

All evaluation experiments for SPCANet were performed on four mainstream datasets: ShanghaiTech Part A, ShanghaiTech Part B, UCF-QNRF, and UCF_CC_50. The Overview of the four datasets is shown in Table 2.

    **ShanghaiTech** (*Zhang et al., 2016*) is a large-scale crowd-counting dataset released by the Shanghai University of Science and Technology in 2016, divided into Part A and Part B, with 1,198 crowd images. The Part A images are from the Internet, with 482 total images split into a training set of 300 and a test set of 182. The images are densely populated and have variable resolution. The Part B images are taken from the streets of Shanghai, and this part contains 716 images, 400 images in the training set, and 316 images in the test set. The targets are sparse. The image resolution remains the same. On the whole, Shanghai tech is very diverse in both scene types and viewpoint transformations, and it is still challenging to realize accurate counting on it.

**Table 3 Comparisons of four Mainstream datasets.** The best results are shown in bold.

| Methods | Shanghai B | | Shanghai A | | UCF-QNRF | | UCF_CC_50 | |
|---|---|---|---|---|---|---|---|---|
| | MAE | MSE | MAE | MSE | MAE | MSE | MAE | MSE |
| MCNN (*Zhang et al., 2016*) (2016) | 26.4 | – | 110.2 | – | – | – | 337.6 | – |
| CSRNet (*Li, Zhang & Chen, 2018*) (2018) | 10.6 | 16 | 68.2 | 115 | 120.3 | 208.5 | 266.1 | 397.5 |
| SANet (*Cao et al., 2018b*) (2018) | 8.4 | 13.6 | 67 | 104.5 | | | 258.4 | 334.9 |
| CAN (*Liu, Salzmann & Fua, 2019*) (2019) | 7.8 | 12.2 | 62.3 | 100 | 107 | 183 | 212.2 | 243.7 |
| LSC-CNN (*Sam et al., 2020*) (2019) | 8.1 | 12.7 | 66.4 | 117 | 120.5 | 218.2 | 225.6 | 302.7 |
| DensityCNN (*Jiang et al., 2020*) (2020) | 9.1 | 16.3 | 63.1 | 106.3 | 101.5 | 186.9 | 244.6 | 341.8 |
| SDANet (*Miao et al., 2020*) (2020) | 7.8 | **10.2** | 63.6 | 101.8 | – | – | 227.6 | 316.4 |
| FusionCount (*Ma, Sanchez & Guha, 2022*) (2022) | **6.9** | 11.8 | 62.2 | 101.2 | – | – | – | – |
| OrdinalEntropy (*Zhang et al., 2023*) (2023) | 9.1 | 14.5 | 65.6 | 105 | – | – | – | – |
| NDConv (*Zhong et al., 2022*) (baseline) | 7.8 | 13.8 | 61.4 | 104.1 | 91.2 | 165.6 | 167.2 | 240.6 |
| SPCANet | 7.3 | 12.5 | **60.9** | **99.9** | **90.8** | **158.7** | **161.1** | **228.8** |

UCF-QNRF (*Idrees et al., 2018*) released by the esteemed University of Florida in 2018, is a comprehensive collection of 1,535 crowd images, meticulously divided into a training set of 1,201 images and a test set of 334 images. This dataset boasts an impressive total of 1.25 million painstakingly detailed head annotations. Compared to the ShanghaiTech dataset, the UCF-QNRF dataset stands out as a more intricate and challenging compilation. It boasts a higher proportion of high-count crowd images, a wider spectrum of diverse viewpoints, and an increased frequency of images depicting varying densities and lighting conditions.

UCF_CC_50 (*Idrees et al., 2013*) is a unique challenge in crowd counting to test the performance of models under crowded crowd images. It has only 50 images with a total of 63,974 head center annotations. These images were mainly collected from FLICKR. The number of heads per image ranges from 94 to 4,543, characterized by small data size and high variance. This data resource plays an irreplaceable role in comprehensively assessing counting methods' performance and efficiency in realistic scenarios with complexity and variability.

## Evaluation and comparison

In the testing and comparison of models, we selected ten models, MCNN (*Zhang et al., 2016*), CSRNet (*Li, Zhang & Chen, 2018*), SANet (*Cao et al., 2018b*), CAN (*Liu, Salzmann & Fua, 2019*), LSC-CNN (*Sam et al., 2020*), DensityCNN (*Jiang et al., 2020*), SDANet (*Miao et al., 2020*), FusionCount (*Ma, Sanchez & Guha, 2022*), OrdinalEntropy (*Zhang et al., 2023*), and NDConv (*Zhong et al., 2022*), and, as mentioned above, the metrics for evaluating the models are the mean absolute error (MAE) and mean squared error (MSE).

Table 3 lists a comparison of SPCANet with 16 other population counting methods, with the best results shown in bold. Compared to the current state-of-the-art methods, we achieved better results.

As shown in Fig. 3, in Shanghai Part B, SPCANet is ranked second, with 0.5 performance improvement and 9.4% model robustness improvement compared to baseline, indicating

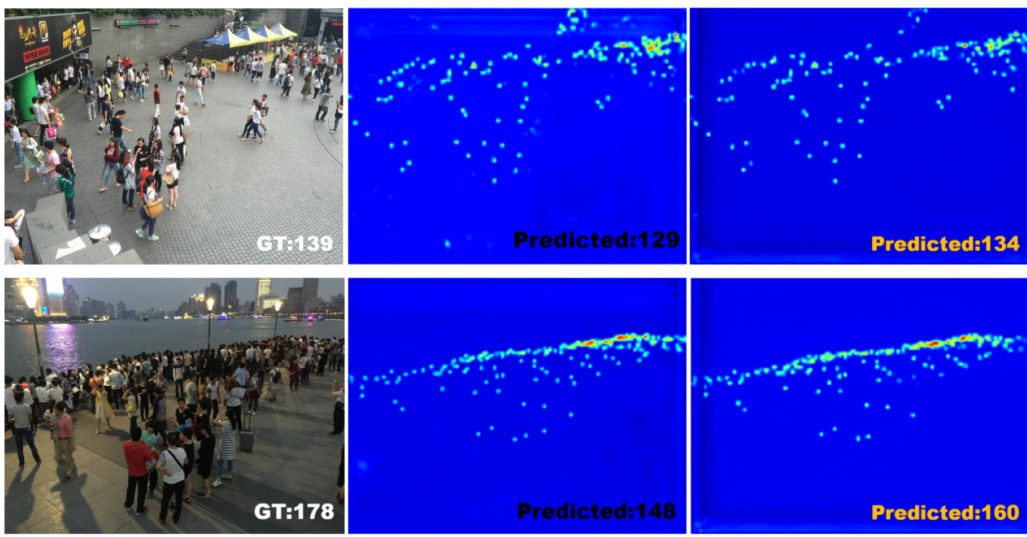

**Figure 3** Visualization of an example from Shanghai Part B dataset.

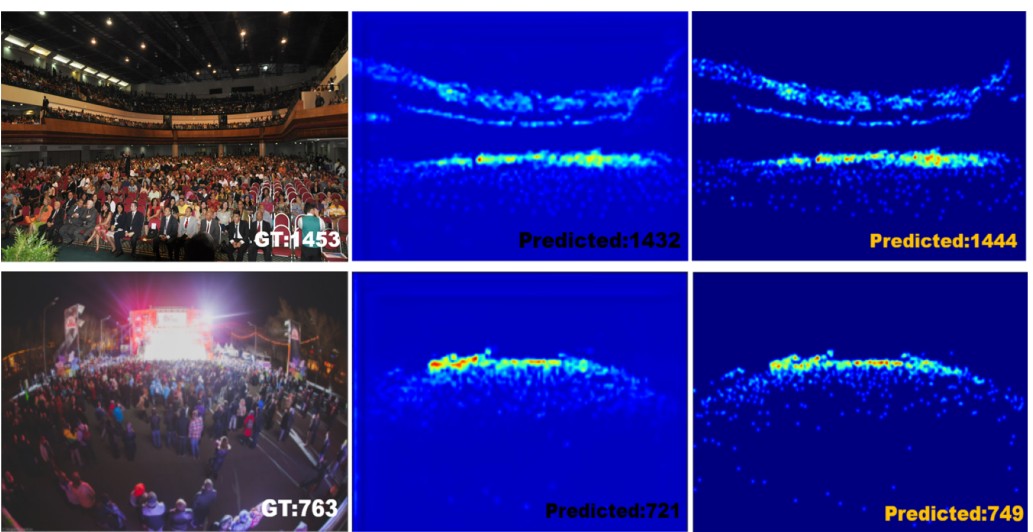

**Figure 4** Visualization of an example from Shanghai Part A dataset.

that SPCANet achieves good results in environments with minor changes in scene perspectives. As shown in Fig. 4, in Shanghai Part A, the performance is improved by 0.5 compared to the baseline, and the MSE is decreased by 4.1%, which shows that the focusing long-distance module we added has achieved good results. Facing the UCF-QNRF dataset, which has a larger volume and more diversified viewpoints, the mae improves by 0.4 compared to the baseline, and the MSE decreases by 4.2%, which indicates that

**Table 4  Influence of the number of strip pooling on Shanghai Part A.** The best results are shown in bold.

| Strip pooling | Baseline (NDConv) | | Ours (SPCANet) | |
|---|---|---|---|---|
| | MAE | MSE | MAE | MSE |
| 1 | **61.4** | 104.1 | 62.2 | 105.3 |
| 2 | 61.9 | **103.8** | **60.9** | **99.9** |
| 3 | 63.7 | 111.2 | 62.9 | 110.7 |
| 4 | 64.6 | 108.3 | 62.6 | 107.9 |

our improvement is more capable of coping with the tasks of larger and more complex scenarios.

SPCANet is better at dealing with dense crowd scene tasks. When facing the 50 images in UCF_CC_50 in extreme environments, our model also improves significantly compared to the baseline, and the MAE is improved by 6.1, which shows the feasibility of long-distance focusing in extreme environments. This is a good improvement in SPCANet's ability to deal with distortion caused by perspective distortion and crowd distribution diversity.

## Ablation study

In this section, the primary purpose of our ablation experiments 1 and 2 is to verify the effect of the number of strip poolings in the network and the effectiveness of the ECA. Experiment 1 discusses the performance increase of the strip pooling module to SPCANet, and experiment 2 discusses the subtle changes that ECA brings to the whole SPCANet when it works with the strip pooling module. Experiments 3 and 4 mainly verify the effects of NDloss and some hyperparameter settings of the model on SPCANet so that the latter can reproduce the experimental results.

**Influence of the number of strip pooling.** NDConv replaces the null convolution in CSRNet with deformable convolution and replaces it with NDConv at each layer of deformable convolution, and we add a strip pooling after NDConv to experiment on Shanghai Part A. As shown in Table 4, with the increase of the number of variability layers, *i.e.,* the number of strip poolings, the performance firstly increases and then decreases from 61.4 to 64.6 for NDConv. The trend of SPCANet is the same as that of NDConv, from 60.9 to 62.9. Moreover, the best result was obtained by adding strip pooling after the two NDConv.

**Effect of ECA on further model improvements.** As mentioned above, efficient channel attention (ECA) can obtain higher model performance at the cost of increasing the number of parameters by a minimal amount. In the experiments on ECA to further improve the model performance, we divided them into three groups of methods: baseline (NDConv), baseline + strip pooling, baseline + strip pooling + ECA, and two by two to form a control. As shown in Table 5, adding strip pooling to NDConv improves the model performance considerably, especially on larger datasets. The MAE decreases from 61.4 to 60.6 on Shanghai A, and the performance on UCF_CC_50 increases by 2.1, which verifies that the introduction of strip pooling can improve the model's counting ability in highly congested

**Table 5  Effect of ECA on further model improvements.** The best results are shown in bold.

| Methods | Shanghai B | | Shanghai A | | UCF_QNRF | | UCF_CC_50 | |
|---|---|---|---|---|---|---|---|---|
| | MAE | MSE | MAE | MSE | MAE | MSE | MAE | MSE |
| baseline | 7.8 | 13.8 | 61.4 | 104.18 | 91.2 | 165.6 | 167.2 | 240.6 |
| baseline+strip pooling | 7.8 | 12.9 | **60.6** | 102.81 | **89.4** | 160.9 | 165.1 | 253.3 |
| baseline+strip pooling+ECA | **7.3** | **12.5** | 60.9 | **99.9** | 90.8 | **158.7** | **161.1** | **228.8** |

**Table 6  Effect of the weight of $\mathcal{L}_{nd}$ on Shanghai tech part A and part B.** The best results are shown in bold.

| Dataset | Metrics | $\lambda$ (super-parameter) | | | |
|---|---|---|---|---|---|
| | | $1e^{-1}$ | $1e^{-2}$ | $1e^{-3}$ | $1e^{-4}$ |
| Shanghai Tech Part A | MAE | 63.1 | 62.7 | **60.9** | 61.4 |
| | MSE | 100.1 | 99.6 | 99.9 | **98.6** |
| Shanghai Tech Part B | MAE | 7.6 | 7.4 | **7.3** | 7.7 |
| | MSE | 14.7 | **12.2** | 12.5 | 12.8 |

and noisy scenarios, as mentioned above. The introduction of ECA enhances the model's performance on most datasets.

It shows that ECA does not exacerbate the complexity of the model, which is also consistent with our previous conclusion. On Shanghai B, the MAE drops from 7.8 to 7.3. Compared to adding only strip pooling, the introduction of ECA makes the model more delicate in dealing with scenarios with sparse crowds. On UCF_CC_50, the performance is improved by 6.1, and the channel attention can also handle the distortion of viewpoints and the diversity of crowd distribution in extreme environments very well. In addition, from the MSE changes, the introduction of ECA dramatically improves the robustness of the model. Shanghai B improves by 9.4%, Shanghai A improves by 4.1%, UCF_QNRF improves by 4.2%, and UCF_CC_50 improves by 4.9%. Although there is a significant improvement from the results of Shanghai A and UCF_QNRF, our model's ability to cope with large datasets is reduced after the introduction of ECA.

**The impact of hyperparameters.** Firstly, as an essential component of the loss function, the effect of the hyperparameter $\lambda$ is investigated. Through ablation experiments, it was verified that the model performs better when the hyperparameter $\lambda$ weight is set to $1e^{-3}$. Experimental data are shown in Table 6. Similarly, Table 7 explores the impact of the optimizer and batch size parameters on performance. It was confirmed that the model performed best when the batch size was set to 8 on UCF-QNRF and 4 on UCF_CC_50 and when the Adam optimizer was used.

## CONCLUSIONS

In this article, SPCANet is presented based on NDConv. A novel spatial pooling operation, strip pooling, is innovatively added to the count-based network. The long and narrow pooling window allows the model to collect the rich global crowd perspective information in the picture with good results. This lightweight module does not cause additional

**Table 7 Effect of optimizer and BatchSize on UCF-QNRF and UCF_CC_50.** The best results are shown in bold.

| Dataset | Metrics | Optimizer & BatchSize | | | | | |
| --- | --- | --- | --- | --- | --- | --- | --- |
| | | Adam | | | SGD | | |
| | | 4 | 8 | 16 | 4 | 8 | 16 |
| UCF-QNRF | MAE | 91.1 | **90.8** | 92.3 | 93.7 | 95.6 | 94.1 |
| | MSE | **157.9** | 158.7 | 161.3 | 165.4 | 165.2 | 167.9 |
| UCF_CC_50 | MAE | **161.1** | 175.2 | 182.4 | 167.2 | 187.2 | 189.5 |
| | MSE | 228.8 | 238.7 | 283.1 | 240.6 | **227.5** | 240.6 |

computational burden. On top of enriching the model perspectives, efficient channel attention (ECA), a mechanism for learning channel attention using a local cross-channel interaction strategy, is added to the model to improve the performance as much as possible at the cost of adding very few parameters.

In the SPCANet, we mainly focus on strip pooling to capture global contextual information. In the future, we can explore methods for multi-scale feature fusion to further enhance the model's adaptability to different crowd densities by extracting and fusing features at various scales. Efficient channel attention (ECA) has demonstrated its effectiveness. Future research could investigate dynamic attention mechanisms to adaptively adjust attention weights based on the features of input images, thus improving the model's robustness and accuracy. We believe that SPCANet can be extended to other similarly challenging counting scenarios and look forward to validating its effectiveness in the future, thereby advancing the development of crowd counting and related fields.

### Funding

The authors received no funding for this work.

### Competing Interests
The authors declare there are no competing interests.

### Author Contributions

- Zhongyuan Yuan conceived and designed the experiments, performed the experiments, analyzed the data, performed the computation work, prepared figures and/or tables, authored or reviewed drafts of the article, and approved the final draft.

### Data Availability
The three external datasets are available at:

- ShanghaiTech: https://github.com/desenzhou/ShanghaiTechDataset
- UCF-QRNF: https://www.crcv.ucf.edu/data/ucf-qnrf
- UCF_CC_50: https://www.crcv.ucf.edu/data/ucf-cc-50/

The datasets, all the pre-trained models, and the source code are all available at Figshare:
- Zhongyuan, Yuan (2024). Datasets for SPCANet. figshare. Dataset. https://doi.org/10.6084/m9.figshare.25360387.v1
- Zhongyuan, Yuan (2024). Pretrained Model for SPCANet. figshare. Journal contribution. https://doi.org/10.6084/m9.figshare.25360606.v1
- Zhongyuan, Yuan (2024). Code for SPCANet. figshare. Journal contribution. https://doi.org/10.6084/m9.figshare.25360624.v1

The code is available at GitHub and Zenodo: https://github.com/YUANBAO425/SPCANet.git.
- YUANBAO425. (2024). YUANBAO425/SPCANet: First release of SPCANet (v1.0.0). Zenodo. https://doi.org/10.5281/zenodo.13264669

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
