# Peer review of "SPCANet: congested crowd counting via strip pooling combined attention network"

_PeerJ Computer Science, doi:10.7717/peerj-cs.2273_

## Round 0.1 · original submission · Major Revisions

Dear authors,
You are advised to critically respond to all comments point by point when preparing a new version of the manuscript and while preparing for the rebuttal letter. Please address all the comments/suggestions provided by the reviewers.

Kind regards,
PCoelho

Reviewer 1 ·

Basic reporting

The paper is clear and precise, using professional English consistently. It includes relevant literature to provide context for the discussion. Additionally, the introduction effectively sets up the background and importance of the research in the broader academic conversation. A new blend of existing methods has led to incremental contributions to the experimental outcomes.

There are minor formatting issues, that should be fixed. It is recommended that the whole paper be carefully reviewed again for minor formatting errors. Examples:
The same sentence is repeated in lines 43-48.
Line 60: The space before reference is missing.
Line 219: space missing after β€œ.”.
Repeating in line 142: β€žIn 2018, Li et al. Li et al. (2018) proposed...β€œ

All appropriate raw data must be available in accordance with PeerJ Data Sharing policy (https://peerj.com/about/policies-and-procedures/#data-materials-sharing). The source code is not provided and should be included before acceptance!

Experimental design

no comment

Validity of the findings

no comment

Cite this review as

Reviewer 2 ·

Basic reporting

The author proposed a network model named SPCANet, which is based on NDConv. The proposed model integrates the ECA module, NDloss, Strip Pooling, and ECA module into one model. However, most of these modules are off-the-shelf and have not undergone any modifications, rendering their direct application lacking in innovativeness.
Although the motivation and theoretical frameworks of each algorithm have been introduced, there is a lack of detailed descriptions of the specific implementation of these algorithms and their integration with existing models. The paper could have further elaborated on the nuances of how these various components work together to enhance the performance of the SPCANet model. Additionally, discussing the challenges encountered during the integration process and how they were overcome would have provided a more comprehensive understanding of the model's development. Furthermore, the author's writing skills require further refinement, as the textual expressions are often not as rigorous and formal as they should be. The language and vocabulary used tend to be superficial and lack depth and impact.

Experimental design

The author designed experiments to evaluate the model. however, it is suggest to provide more implementation details, such as pseudocode and parameter settings, which would greatly assist readers in understanding your methodology and reproducing the experimental results. Additionally, the comparison with the latest state-of-the-art (SOTA) methods is limited, and the majority of the methods discussed in the article tend to be outdated, lacking in-depth analysis.

Validity of the findings

The proposed model and implementation outlined in this paper are feasible, with the author providing relevant code, clearly indicating the datasets utilized, and comparing the experimental results. From an engineering perspective, the overall methodology is well-articulated and comprehensible.

Cite this review as

Reviewer 3 ·

Basic reporting

All comments have been added in detail to the last section.

Experimental design

All comments have been added in detail to the last section.

Validity of the findings

All comments have been added in detail to the last section.

Additional comments

Review Report for PeerJ Computer Science
(SPCANet: Congested crowd counting via strip pooling combined attention network)

1. Within the scope of the study, a new deep learning based network model has been proposed by performing various operations on different datasets for crowd counting operations.

2. In the introduction section, the importance of rudeness counting, counting methods, crowd shelter, perspective distortion, inconsistent distribution and the main contributions of the study have been mentioned. In this section, both the importance of the subject and the main contributions have been explained clearly.

3. In the Related works section, regression-based, CNN-based and detection-based methods have been mentioned regarding crowd counting studies in the literature. In this section, in order to emphasize the place of the subject in the literature and to highlight the originality of the study; it is suggested to add a literature table consisting of columns such as "used dataset, model, pros, cons, results" in these studies in the literature related to crowd counting.

4. In the Methods section, pooling, loss and efficient channel attention, along with the proposed network architecture, and the developments made accordingly have been mentioned. When the model architecture is examined, it is observed that significant developments have been made, and it is noticeable that VGG16 is used in the backbone network. When the literature is examined, there are many different deep learning-based backbone networks that can be used in this context. Please explain in detail why this one is preferred and/or why experiments with different models are not conducted.

5. In the Experiment section, various hyperparameter values (such as optimizer, learning rate, batch size) used in the study regarding the setting are briefly mentioned. How these parameters are determined and the values used are very important in terms of problem solutions. For this reason, it should be clearly stated on what basis these parameter values and parameter types are selected and whether different experiments are conducted.

6. It is stated that a total of four different datasets, UCF CC 50, UCF-QNRF and two different ShanghaiTech, are used as datasets. In terms of dataset diversity and the application of the proposed model accordingly, both the amount of dataset and the selection of datasets are sufficient. In this section, it is recommended to add a table consisting of differences such as the total amount of data, training amount, test amount, etc. for each dataset for comparison.

7. The mean squared error and mean absolute error metrics, which are the evaluation metrics required for the analysis of the results, were obtained for each dataset using the proposed model and compared with other similar studies in the literature. Both the type of metrics, the analysis of the results, and the comparisons of the results of the proposed model with other models are sufficient.

8. Although the conclusion section is basically sufficient, since a new model is proposed within the scope of the study, it is recommended to add a detailed future works section to this section to potentially inspire future application studies.

As a result, although the study proposes a model that has the potential to contribute to the literature in relation to crowd counting, it is recommended to pay attention to the sections listed above.

Cite this review as

---

## Round 0.2 · accepted · Accept

Dear author, we are pleased to verify that you meet the reviewer's valuable feedback to improve your research.

Thank you for considering PeerJ Computer Science and submitting your work.

Kind regards,
PCoelho

Reviewer 2 ·

Basic reporting

This paper has been revised in detail based on the comments of the reviewers. The improvements in logic and language is quite obvious. The expression of main parts are also more fluent. However, the reference citation and main text of the manuscript are mixed together without using parentheses to distinguish, making it very difficult to read. It is recommended that the author carefully revise it again.

Experimental design

The experimental section conducted a detailed comparison to demonstrate the performance of the method as much as possible.

Validity of the findings

The innovation of the method is still relatively weak, and its impact on the community seems trivial.

Cite this review as

Reviewer 3 ·

Basic reporting

All comments have been added in detail to the last section.

Experimental design

All comments have been added in detail to the last section.

Validity of the findings

All comments have been added in detail to the last section.

Additional comments

Review Report for PeerJ Computer Science
(SPCANet: Congested crowd counting via strip pooling combined attention network)

Thanks for the revision of this manuscript according to my comments. Some responses to the reviewer comments are limited, but overall, since the changes made in the paper are sufficient and the paper has the potential to contribute to the literature, I recommend that this research paper be accepted. I wish the authors success in their future work. Best regards.

Cite this review as